# Topical Imiquimod in Primary Cutaneous Extramammary Paget’s Disease: A Systematic Review

**DOI:** 10.3390/cancers15235665

**Published:** 2023-11-30

**Authors:** Fátima Mayo-Martínez, Ruggero Moro, David Millán-Esteban, Elisa Ríos-Viñuela, Iker Javier Bautista, Eduardo Nagore, Onofre Sanmartín, Beatriz Llombart

**Affiliations:** 1Departments of Dermatology, Fundación Instituto Valenciano de Oncología, 46009 Valencia, Spain; dmillan@fivo.org (D.M.-E.); eduardo.nagore.e@gmail.com (E.N.); onofre.sanmartin@ucv.es (O.S.); 2Escuela de Doctorado, Universidad Católica de Valencia San Vicente Mártir, 46001 Valencia, Spainelisariosvi@hotmail.com (E.R.-V.); 3Tekderma Instituto Dermatológico, Hospital Vithas Valencia 9 de Octubre, 46015 Valencia, Spain; 4Department of Physiotherapy, Catholic University of Valencia, 46001 Valencia, Spain; ij.bautista@ucv.es; 5Institute of Sport, Nursing and Allied Health, University of Chichester, Chichester PO19 6PE, UK

**Keywords:** imiquimod, topical immunotherapy, extramammary Paget’s disease, vulvar Paget’s disease, extramammary Paget disease, scrotal Paget’s disease, perianal Paget’s disease, cutaneous Paget’s disease, treatment

## Abstract

**Simple Summary:**

The main treatment for extramammary Paget’s disease (EMPD) is still surgery, but this neoplasm usually spreads beyond the clinical surroundings of the lesion, and achieving histopathologically free margins can be challenging and requires mutilating surgery. Moreover, the recurrence rate of the disease is high, reflecting its multifocal nature. Topical immunotherapy could be an alternative treatment for EMPD. The aim of this systematic review was to assess the effectiveness of topical imiquimod in the clinical response of EMPD patients and to describe the management of topical imiquimod in EMPD. Learning about non-surgical treatments such as topical imiquimod can help clinicians manage EMPD and find a balance between disease resolution and treatment morbidity.

**Abstract:**

Extramammary Paget’s disease (EMPD) is subclinical in extent and multifocal in nature. There is no global consensus for treatment, so its management represents a challenge in clinical practice. Therefore, we conducted a systematic review through the main electronic databases to assess the effectiveness of topical imiquimod in cutaneous EMPD and to discuss its management. Finally, 24 studies involving a total of 233 EMPD patients treated with topical imiquimod were selected. The topical imiquimod response rate was 67%, and the complete response (CR) rate was 48%. Patients were treated with a three–four times a week regimen in most cases, ranging between 2 to 52 weeks. In addition, imiquimod was applied as an adjunctive treatment in 21 patients, achieving a CR rate of 71%. Consequently, imiquimod therapy could achieve a good response ratio as a first-line treatment, as adjuvant and neo-adjuvant therapy, and as a treatment for recurrent disease. The heterogeneity between studies and the lack of a control arm made it impossible to conduct a meta-analysis. To improve the quality of evidence on EMPD, multicenter studies are essential to collect a larger number of patients and, consequently, obtain high-quality evidence to standardize treatment. The Prospero registration number is CRD42023447443.

## 1. Introduction

Extramammary Paget’s disease (EMPD) is a rare cutaneous malignant neoplasm affecting the apocrine gland-bearing skin [1] (see Figure 1). The most frequently affected anatomical sites, in order of frequency, are the vulva, scrotum–penis, perianal region, and axilla [2,3,4]. Given its rarity, the true incidence of EMPD remains unclear, although it has been estimated to be between 0.11–0.7 per 100,000 person-years [5,6,7,8,9].

The distinction between primary and secondary disease is important due to its prognostic implications. Primary cutaneous EMPD (EMPD) originates in the epidermis and can be classified as in situ or intraepithelial disease (the most common form), invasive neoplasm (with locoregional or distant metastatic potential), or associated with an underlying adenocarcinoma of a skin appendage or a subcutaneous gland. Secondary EMPD (S-EMPD) has a non-cutaneous origin; namely, the cutaneous involvement is secondary to a metastatic or epidermotropic spread of malignant cells from anorectal, urothelial, or other adenocarcinomas. This underlying adenocarcinoma can be synchronous or asynchronous [9,10,11,12].

The main treatment for EMPD is still surgery, but EMPD usually spreads beyond the clinical surroundings of the lesion, making it difficult to define the lesion borders. For this reason, achieving histopathological-free margins can be challenging and require mutilating surgery. Moreover, the recurrence rate of the disease is high, reflecting its common subclinical extension and multifocal nature [4,9,10,13,14]. In parallel, non-surgical treatments have been implemented, like photodynamic therapy (PDT), topical chemotherapy (bleomycin, 5-fluorouracil, and ingenol mebutate), topical immunotherapy (imiquimod), laser ablation, and radiotherapy (RT) [6,9,15,16]. Given the wide therapeutic range, several authors have worked to shed light on the management of EMPD. All of them concluded that there was a lack of high-quality evidence to establish a gold standard treatment for EMPD patients [4,6,9,10,17,18,19].

Since Zampogna et al. reported in 2002 the first two cases of EMPD treated with topical imiquimod, this cream has gained attention as an off-label, topical, functional preserver, and tissue-sparing treatment [20]. Imiquimod is a toll-like receptor agonist that stimulates the innate natural local immunity and the adaptive cell-mediated TH-1 immune response, as well as inhibiting the TH-2 pathway, overexpressed in skin cancer [20,21,22]. Thus, the immune system attacks the intraepidermal spread of Paget cells, both the visible skin lesion and the subclinical disease [8]. Furthermore, despite the fact that EMPD can be very extensive and multifocal, in the vast majority of cases, it is an in-situ neoplasm. This supports the use of topical imiquimod as a valid therapeutic approach in most cases (see Figure 2). Recently, a statement issued by a board of experts recommended the use of imiquimod in vulvar Paget disease [16]. Furthermore, the prognosis of non-invasive and microinvasive EMPD is outstanding, with a 5-year survival of 90 to 100% [10,23,24].

The aim of this systematic review was to assess the effectiveness of topical immunotherapy on clinical response in EMPD patients. The second aim of this study was to describe the management of topical imiquimod in EMPD.

## 2. Materials and Methods

### 2.1. Study Design

The design of this systematic review was performed according to the Reporting Items for Systematic Reviews and Meta-analysis (PRISMA) guidelines (see Appendix A) [25]. The protocol was registered on the International Prospective Register of Systematic Reviews PROSPERO before searches, data extraction, and data analysis (CRD42023447443).

### 2.2. Eligibility Criteria

#### Inclusion and Exclusion Criteria

Due to the limited number of studies on this topic, flexible eligibility criteria were applied. We included all interventional studies, randomized and non-randomized. We included studies conducted in both men and women, regardless of age, without restriction by location. In order to be included, studies must use topical imiquimod to treat patients with EMPD. We considered both studies in which patients were treated with imiquimod alone, as well as studies in which patients also received other treatments (5-fluorouracil, bleomycin, ingenol mebutate, photodynamic therapy, laser, radiotherapy, or surgery).

Studies that had been conducted in patients with cutaneous extramammary Paget’s disease secondary to an underlying neoplasm or in patients with advanced disease were excluded.

Conference proceedings, abstracts, and other unpublished studies were excluded. Narrative review, systematic review, and meta-analysis were also excluded. Single-case reports were also excluded to reduce the positive-outcome publication bias. Finally, publications with incomplete information were also excluded.

Only articles written in English, Spanish, or French were considered for this systematic review.

### 2.3. Search Strategy

The primary search focused on studies reporting on the effect of topical imiquimod on EMPDc lesions and was performed up to July 2023. Four independent reviewers (F.M.-M), E.R.-V., O.S., and B.L.) performed the electronic search through OVID MEDLINE/Pubmed, Web of Science, the Cochrane Library, and OVID Embase databases. No date restriction was placed on this search. We applied forward and backward snowballing of the identified relevant papers and adapted the search in case of additional relevant studies. The PICO strategy was used to develop the search criteria for the electronic databases. This consisted of terms for EMPDc and topical imiquimod. Key search terms included “extramammary Paget’s disease”, “extramammary Paget disease”, “vulvar Paget’s Disease”, “scrotal Paget’s Disease”, “perianal Paget’s disease”, “cutaneous Paget’s disease”, “perineum Paget’s disease”, “inguinal Paget’s Disease”, and EMPD in combination with “topical immunotherapy” and imiquimod.

The primary search stream was built up with Pubmed: (“topical immunotherapy” [All Fields] OR (“imiquimod” [MeSH Terms] OR “imiquimod” [All Fields])) AND (“extramammary Paget’s disease” [All Fields] OR “Extramammary Paget Disease” [All Fields] OR “vulvar Paget’s Disease” [All Fields] OR “scrotal Paget’s Disease” [All Fields] OR “perianal Paget’s disease” [All Fields] OR “cutaneous Paget’s disease” [All Fields] OR (“perineum” [MeSH Terms] OR “perineum” [All Fields] OR “perineums” [All Fields]) AND (“osteitis deformans” [MeSH Terms] OR (“osteitis” [All Fields] AND “deformans” [All Fields]) OR “osteitis deformans” [All Fields] OR (“paget s” [All Fields] AND “disease” [All Fields]) OR “paget s disease” [All Fields])) OR ((“groin” [MeSH Terms] OR “groin” [All Fields] OR “inguinal” [All Fields] OR “inguinally” [All Fields]) AND (“osteitis deformans” [MeSH Terms] OR (“osteitis” [All Fields] AND “deformans” [All Fields]) OR “osteitis deformans” [All Fields] OR (“paget s” [All Fields] AND “disease” [All Fields]) OR “paget s disease” [All Fields])) OR “EMPD” [All Fields]). The search strings used for other databases were adapted using the Polyglot Search Translator Tool (https://sr-accelerator.com/#/polyglot, accessed on 25 August 2023) [26].

### 2.4. Data Extraction

Three independent reviewers (F.M.-M., E.R.-V., and B.L.) examined the titles and abstracts of all studies initially identified. Articles fulfilling the inclusion/exclusion criteria were selected, and full texts were retrieved. Cross-referenced studies identified from searched articles were also evaluated to integrate the literature search. Two authors (F.M.-M. and B.L.) independently checked the full text, excluded articles that were arguably not eligible, extracted the data, and performed the quality assessments. In the case of overlapping studies, the most complete manuscript was selected. Finally, a consensus was reached with the collaboration of all authors. The collected data from the selected studies were as follows: first author, year of publication, study design, number of participants, number of patients with EMPD treated with imiquimod cream, sex, age at diagnosis, race, number of lesion(s), location of lesion(s), tumor size (cm), tumor stage (TNM proposed by Ohara et al.) [27], duration of EMPD (months), topical imiquimod treatment characteristics (initiation of treatment [first line topical imiquimod treatment or use in other lines], dosage, duration, overall days of application, retreatments needed, treatment response [complete or partial response, stable or progressive disease], and side effects), other treatments used, follow-up period (years) and disease-free survival (months), and conflicts of interest (was expressed as “none declared” when the authors had no conflicts of interest, “declared” when the authors have conflicts of interest, and “not available” when the authors do not declare if they have conflicts of interest or not). Missing data were expected, and study investigators would not be contacted for any unreported data/additional details.

The level of evidence (LE) of each article was determined based on the Oxford 2011 Levels of Evidence and included in the extraction data table [28].

### 2.5. Risk of Bias

Two researchers (F.M.-M. and B.L.) independently assessed the methodological quality of the selected studies using the Joanna Briggs Institute (JBI) Critical Appraisal Tools for case reports, case series, or quasi-experimental studies (according to the type of paper) [29]. The observational trials and the cohort studies were considered intervention studies, and thus, the JBI critical appraisal tool for quasi-experimental studies was applied. In case of disagreement between the scores provided, a consensus was reached among all authors.

### 2.6. Statistical Analysis

A narrative synthesis and construction of descriptive summary tables were performed for the included studies. Data from treatment schedules, outcomes of the patient, follow-up, and side effects were analyzed from a descriptive-manner point of view. The ratio outcomes were described by median, mean, as a measure of central tendency, and range or standard deviation, as a measure of uncertainty. Categorical data were described by frequency and percentage. When necessary, for the description of some data, percentages related to the number of patients or studies for which these specific data were available were reported.

## 3. Results

### 3.1. Study Selection

Figure 3 shows the flow chart with the different phases of the systematic literature search and the selection of studies included in this review. The number of search results was 743 records. After the elimination of duplicates (589), another 119 studies were excluded based on the title and/or abstract, and finally, 10 more studies were excluded based on full-text assessment. Hence, the study selection resulted in a total of 24 relevant articles included in the present review on EMPD and topical imiquimod.

### 3.2. Study Quality and Bias Results

According to the Oxford 2011 Levels of Evidence guidelines, most of the included studies were level 4 (19/24), followed by level 3 (4/24) and level 2 (1/24) (see Appendix A). The JBI appraisal tool scores of included studies are reported in Appendix A. Regarding the quasi-experimental studies, none of the five studies included had a control group or multiple measurements of the outcome (pre and post) (items 4 and 5 of JBI score). Regarding the 10 case series studies, only two studies conducted statistical analysis. Finally, concerning the case reports, nine articles fulfilled all the JBI items except item 6 (6/9) and item 7 (8/9).

### 3.3. Study and Population Characteristics

The main characteristics of the included studies are summarized in Appendix A. Among the 24 included studies, 3 were observational trials [30,31,32], and 2 were cohort studies [6,33]; both were considered intervention studies, therefore being quasi-experimental designs. In addition, there were 10 case series [1,3,5,14,15,34,35,36,37,38], and 9 were case reports with a minimum of two participants each [20,21,39,40,41,42,43,44]. No randomized controlled studies were found. The 24 selected studies involved a total of 233 EMPD patients treated with topical imiquimod, ranging from a minimum of 2 patients in case reports to a maximum of 55 patients in the largest cohort study. Demographics and features of the EMPD lesions are summarized in Appendix A. The number of lesions was not reported in eight articles. The tumor size and the duration of EMPD were not reported in 17 and 11 articles, respectively. All studies included non-invasive neoplasms, with the exception of Borella et al., 2022 [6], who also included nine microinvasive tumors.

### 3.4. Treatment Characteristics

The overall treatment characteristics are summarized in Table 1. Out of 233 patients, 112 achieved complete clinical clearance (CR), which represents 48% of the patients, with a mean and standard deviation of 61% ± 35%. Partial clinical clearance (PR) was observed in 45 of the 233 patients, which represents 19% of the patients, with a mean and standard deviation of 23% ± 27%. Sixty-six of the 233 patients were reported to show stable disease (28%), with a mean and standard deviation of 13% ± 29%, and one patient reported a progressing EMPD. Therefore, 67% of patients responded to imiquimod treatment, while 29% did not. All were non-invasive EMPD except for nine microinvasive tumors. Five of these microinvasive tumors achieved a CR. The clinical outcome is detailed in Table 2. After imiquimod treatment, a histopathological analysis was performed in 99 of the 233 patients. Histological clearance of EMPD was observed in 79 of 99 patients (80%) who underwent a biopsy. There was a remnant of lesion in 20 biopsies (20%).

The imiquimod treatment regimen ranged from one to seven times per week. In some studies, the regimen was described in general terms, for example, “two or three times per week”, “every other night”, or “schedules varied from one to five times per week”. Likewise, subjects were treated with a three–four times weekly regimen in most cases (174 of 241 patients), followed by two times a week (55 of 241 patients). It should be noted that sometimes, the same patient performed different regimens. In fact, 24 patients did not have this information available. The duration of treatment ranged between 2 to 400 weeks. Liau et al. reported a strikingly longer treatment duration than other authors. If we exclude their data, the duration of treatment ranged from 2 to 52 weeks. Overall days of application were calculated by multiplying the average number of weekly applications with the number of weeks of treatment duration, which ranged from 12 to 56 applications (or 12 to 1200, considering the work of Liau et al.) during the total treatment period. Moreover, even in individual cases, the number of applications per week varied throughout the treatment period.

Information about treatments used as first line was available for 209 patients (see Table 3). Topical imiquimod was the treatment of first choice for 112 patients (54%). Imiquimod was used for recurrent or persistent disease for 97 patients (46%). Surgery was the first line of treatment in 82 patients (42%), whereas laser, PDT, other topical treatments (5-FU, ingenol mebutate), or electrodessication were less common (five, four, five, and one patient, respectively). Imiquimod was applied as adjunctive treatment in 21 patients. Thus, of these 21 patients, 10 patients underwent surgery and topical imiquimod postoperatively, 4 patients were treated with PDT followed by imiquimod, one patient was treated with imiquimod and laser, and 6 patients used imiquimod and another topical treatment. Fifteen (71%) of these 21 patients achieved CR.

The most frequent side effects during imiquimod treatment were local, highlighting erythema, local tenderness, irritation, discomfort, swelling, and erosions, reported in 17 publications. Systemic symptoms like flu-like syndrome, nausea, and vomiting were reported in four articles. Hypopigmentation after imiquimod treatment was described in six patients. Dose reduction or dropout during imiquimod treatment due to side effects was reported in 26 of 173 patients (15%).

The follow-up period ranged from 0 to 37.5 years. In 23 of 200 patients (12%), a recurrence was detected after a 3 to 46-month follow-up period (recurrence data were not available for 33 patients).

## 4. Discussion

The existing research on the effectiveness of topical imiquimod for treating extramammary EMPD patients lacks high-quality evidence. To the best of our knowledge, there were no studies with a control group to compare imiquimod cream with other treatment alternatives like surgery, laser, radiotherapy, or other topical treatments. Consequently, there were no standardized, evidence-based guidelines or consensus on the optimal management. The following systematic review seeks to gather and analyze the best available evidence regarding EMPD and the use of imiquimod treatment, focusing on the clinical outcome and management of this topical immunotherapy. Due to the great heterogeneity of the studies, the population, and the differences in data reporting, it was not possible to perform a meta-analysis.

Based on the 24 included studies, topical imiquimod is an effective treatment for EMPD, with a response rate of 67%. The CR rate was lower (48%) than those reported by two previous systematic reviews (71–73%) [46,47]. It must be taken into account that these two studies were performed in vulvar EMPD, whereas we included different locations, such as the systematic review of all non-surgical treatments by Snast et al., which showed a CR rate (54%) more similar to ours [18]. The perianal area had been associated with a lower probability of response [6]. Furthermore, the systematic review by Machida et al. and Dogan et al. were mainly based on case reports and a small case series, therefore being affected by the positive-outcome publication bias. Analyzing this framing, we observed a high standard deviation (SD) of response (35% for CR, 27% for PR, and 29% for SD), likely due to the weight of studies with low sample size (case series and case report) showing higher response rates versus studies with larger sample size (observational trials and cohort studies) where the complete response rate decreased. To confirm CR, the majority of the authors performed a biopsy. Note that negative biopsies had a limited value given the multicentricity and irregular shape of EMPD, which might have been due to sampling error [14,40]. This would explain the high rate of recurrence after response. To minimize the risk of false negatives and to delimit the remaining disease, a post-treatment mapping can be performed, whether there is a response or not [32]. An absence of data on tumor size had been noted, being unable to correlate treatment outcome with EMPD size in this systematic review. This may have been due to the difficulty derived from the anatomical locations of the disease, with continuity solutions, the extent of the lesions, and their multifocal character. However, it were data worth considering, as larger lesions responded worse [5,6].

Different treatment schedules have been employed; however, there is still no standardized management of topical imiquimod. Our study shows that the preferred regimen was three–four times weekly; this application frequency was related to a greater response [3,6]. It seems that increasing the treatment frequency did not improve CR rates but was predisposed to potential side effects, which may require reducing the dose [46]. The treatment duration was also a matter of debate; while some authors argued that the response was time-dependent and even proposed a 6-month duration [3,5,36,46], others observed that a higher frequency was more important than the duration of the therapy [6]. Maybe the focus should be achieving the greatest local inflammatory reaction, albeit bearable for the patient, which ensures the efficacy of the treatment, as suggested by Serra et al. for actinic keratosis [48]. In The Paget Trial, they observed that all three patients who stopped the treatment because of the side effects had CR [31]. Nevertheless, wider prospective studies and clinical trials are needed to determine the best topical imiquimod regimen for EMPD.

Despite the fact that this review was focused on imiquimod treatment, it was noteworthy that many of the patients had received surgical treatment as the first line (42% of the patients), reflecting that surgery remained the main treatment in the EMPD [10]. However, as we have seen, the majority of the EMPD patients responded to topical imiquimod, being a more conservative initial option [16]. Patients with extensive clinical or subclinical disease may benefit from topical treatment to avoid mutilating surgery. Blind scouting biopsies can help assess subclinical extension but provide only focal information and can have false-negative results [4,49]. The clock mapping seemed to be a useful tool to predict the invasiveness and subclinical extension of EMPD and, consequently, to decide the best initial treatment for our patients [50]. Some studies showed that imiquimod was slightly less effective in recurrent disease than in naive EMPD [6,46], whereas previous authors showed a high rate of clinical response in both scenarios [14,30,36]. Overall, imiquimod was useful both as a first-time therapy and for recurrent disease, being a good tool to treat successive recurrences [30]. In the recurrence of a disease in which imiquimod had been used previously, the treatment could be repeated [51]. Also, we observed that imiquimod was effective as an adjuvant treatment and was especially interesting when surgical resection margins were positive to avoid repeated and mutilating surgeries [1,4,9,47]. On the other hand, an important number of patients achieved partial response. Thus, imiquimod treatment might be used as a neo-adjuvant therapy, enabling more cosmetic and functional surgeries [9,36]. However, Choi et al. observed that initial topical therapies were associated with a higher number of stages of Mohs surgery and an increased recurrence rate [1].

Imiquimod is an agonist for toll-like receptors TLR 7 and TLR 8, resulting in cytokine release and upregulation of NK-cell activity, killer T-cell action, and polyclonal activation of lymphocyte B [21,22,30]. It leads to inhibition of cell proliferation, apoptosis, activation of plasmacytoid dendritic cells, inhibition of tumoral angiogenesis, and increase in intratumoral T lymphocytes [6,21,22,40]. As an immunomodulatory agent that induces tumoral cell apoptosis, local and systemic side effects due to inflammatory activity are expected. In most cases, local symptoms were mild-to-moderate, mainly erythema, irritation, and erosion. Systemic symptoms were uncommon. Only 15% of the patients reduced the schedule or dropped out due to side effects during imiquimod treatment. To reduce side effects, patients could use topical anesthetics and analgesics [31]. Overall, imiquimod is a well-tolerated treatment for EMPD patients. Otherwise, patients with a low clinical and inflammatory response could be retreated with a combination of imiquimod and tazarotene; it has been suggested that adding a retinoid might induce a more potent inflammatory response by enhancing drug penetration [52]. The immune stimulation of imiquimod leads the immune system to attack the intraepidermal spread of Paget cells, even though its effectiveness in micro-invasive (depth of invasion <1 mm) EMPD had also been proven, without recurrence or progression to an invasive disease during follow-up [6]. Beforehand, it was found that non-invasive disease and micro-invasive disease had similar survival rates and good prognosis [13,23,24].

We found a recurrence rate of 12% in a period of up to 46-month follow-up. Thus, the recurrence could appear after a long duration of CR, emphasizing the need for a long follow-up in our patients [4,32,51].

## 5. Conclusions

As far as we know, this is the first systematic review focused on the topical imiquimod treatment for EMPD, regardless of the anatomical area involved. The strength of this study is that it is a systematic review of the literature that includes the latest observational trials and cohort studies with a relatively large sample size. However, it had clear limitations. First, there was a great heterogeneity between the studies; the description of the imiquimod treatment, outcome, recurrence, and follow-up varied between different papers, which made data extraction difficult. This revealed the need for homogenous data reporting methods. Second, all studies lacked a control group. Publication bias and sample-sized bias may also exist. All the above made it challenging to perform a meta-analytical study that reinforced the available evidence.

To conclude, our systematic review emphasizes imiquimod therapy as an effective management option for EMPD, a mainly intraepidermal disease with a favorable prognosis. This non-surgical treatment could achieve a good response ratio as first-line treatment, as adjuvant and neo-adjuvant therapy, and as a treatment for recurrent disease. Accordingly, it can improve the quality of life of our patients, avoiding the need for mutilating surgeries with aesthetic and functional sequelae. Finally, long-term follow-up after imiquimod treatment is essential to detect recurrences in EMPD patients. To improve the quality of evidence on this rare neoplasm, international collaboration is essential to collect a larger number of patients and, consequently, to obtain high-quality evidence to standardize treatment.

## Figures and Tables

**Figure 1 cancers-15-05665-f001:**
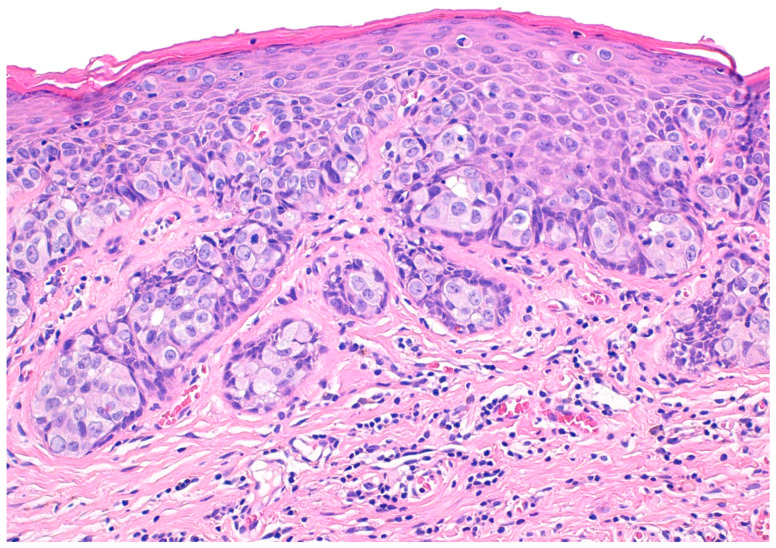
Histopathologic features of extramammary Paget’s disease. Paget’s cells proliferate within the epidermis mainly as isolated units, associating a nested pattern in some areas. Paget’s cells are large, with ample, pale, and finely granular cytoplasm and round and pleomorphic nuclei.

**Figure 2 cancers-15-05665-f002:**
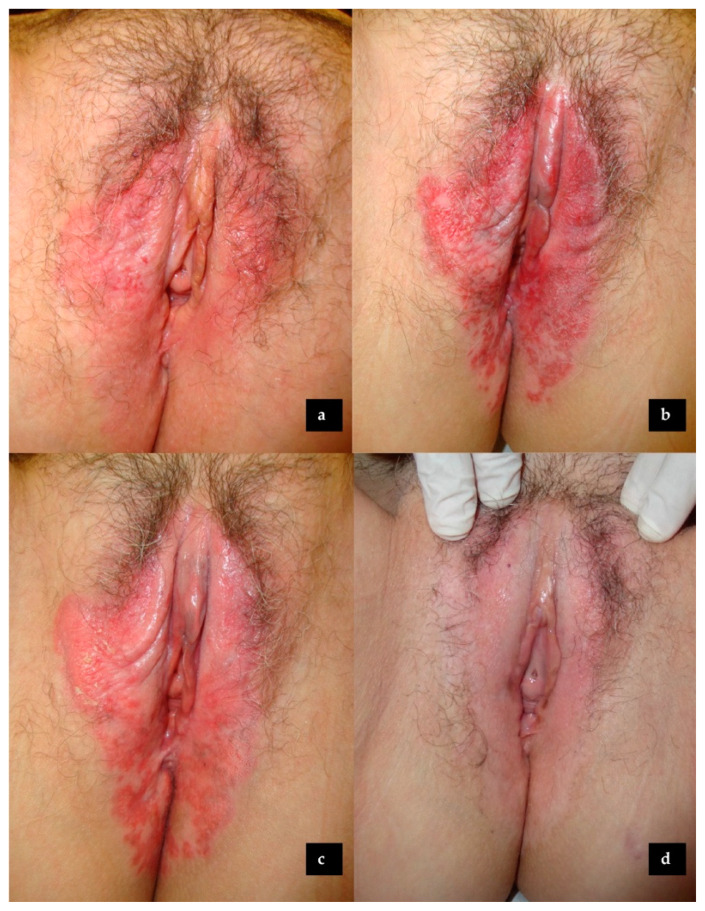
Extramammary Paget’s disease of the vulva treated with topical imiquimod: (**a**) Before treatment; (**b**) After 9 weeks (27 applications) of topical imiquimod; (**c**) After 15 weeks (47 applications) of topical imiquimod; (**d**) Six months after the end of topical imiquimod therapy.

**Figure 3 cancers-15-05665-f003:**
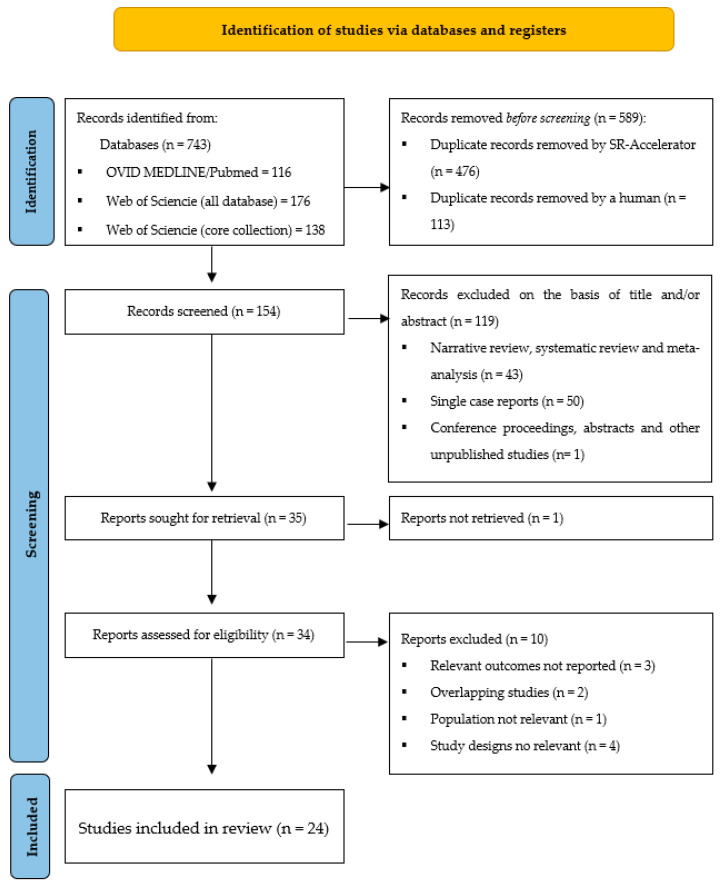
PRISMA flow chart. Provides details on the literature search.

**Table 1 cancers-15-05665-t001:** Overall treatment characteristics.

N°	IMQ Initiation: (1) First Line, (2) Adjunctive Treatment, (3) Other Lines	Order of IMQ Cream in the Sequence of Treatments	Dosage: Times/ Week	Dose Reduction (p; %): Its Reason	Individual Treatment DURATION (Weeks)		Overall Treatment Duration (Weeks)		Overall Days of Application (Days)		Therapeutic Outcome: (1) CR, (2) PR, (3) SD, (4) PD	Clinical Response	Histopathological Response		Recurrence: Yes/No; (Period: Month)	IMQ Retreatments Needed	Side Effects	Follow-Up Period (Years)		Other Treatments Used (Line)	
					Md (M)	Range (SD)	Md (M)	Range (SD)	Md (M)	Range (SD)								Md (M)	Range (SD)		
1 [19]	3	2	7 (1st)/3 (2nd)	Yes: side effects	1 (1st)/16 (2nd)		17		55		1	1 RC	1 RC		No	No	erosion, erythema, and flu-like symptoms; hypopigmentation;	1		electrodesiccation and curettage (1st line)	
	1	1	3	No	7.5		7.5		23		1	1 RC	1 RC		No	No	burning, erythema, nausea, and vomiting; hypopigmentation;	0.5		No	
2 [20]	1	1	5 (1st)/3 (2nd)/2 (3rd)	Yes: side effect (1st)/NR (2nd)	4 (1st)/4 (2nd)/8 (3rd)		16		48		1	1 RC	3 RC		no	no	irritation, erythema; hypopigmentation;	1		No	
	1	1	5 (1st)/3 (2nd	Yes: side effect	4 (1st)/4 (2nd)		8		32		2	1 RP	1 PR		NA	NA	irritation; hypopigmentation;	NR		No	
3 [36]	3	2	3 (1st)/7 (2nd)/4 (3rd)	Yes: side effect (1st)/NR (2nd)	8 (1st)/4 (2nd)/11 (3rd)		23		71		1	1 RC	1 RC		no	no	erosion, erythema	1		Surgery (1st line)	
	3	2	7 (1st)/3 (2nd)	YES: side effects and improvement	7 (1st)/5 (2nd)		12		64		1	1 RC	1 RC		no	no	erosion	0.5		Surgery (1st line)	
4 [38]	2	2	3	No	12		12		36		1	1 RC	NR		no	no	no	0.5		Surgery (1st line)	
	2	2	3	No	12		12		36		1	1 RC	NR		no	no	no	0.3		Surgery (1st line)	
5 [32]	1	1	5	No	24		24		120		2	1 RP	1 PR		NA	NA	no	1		PDT (2nd line)	
6 [5]	1 (8p)/3 (4p)	1 (8p) 2 (3p)/3 (1p)	3	No	5.5	2–12	5.5	2–12	16.5		6–36	1 (6p), 2 (6p)	6 RC/6 RP		NR	No	No	Painful inflammation	2		surgery (1st line in 3p, 1st and 2nd line in 1p)
7 [34]	1 (3p)	1 (3p)	7 (1st)/3 (2nd)	Yes: therapeutic schedule	3 (1st)/3 (2nd)		6		30		1 (6p)	3 RC	3 RC		No: 3p	No (3p)	irritation and tenderness (2p)	4.5 (3p)		No	
8 [42]	1	1	3	No	24		24		72		3	1 SD	NR		NA	NA	Inflammatory reaction	NR		RT (2nd line)	
	1	1	3	No	16		16		48		3	1 SD	NR							RT (2nd line)	
9 [31]	1(1p)/3(3p)	1(1p)/2(3p)	3 (4p 1st)/2(1p 2nd)	yes: side effects (2p)	4 (1p)/16 (1p)/34 (1p)/52 (1p)		25	4–52	59		12–156	1 (1p), 2(2p), 4 (1p)	1 RC, 2 RP, 1 SD		1 RC	Yes: 1p (4)	Yes (1p)	irritation and tenderness (4)	1 (1p), 4 (1p), NR (2p)	surgery (1st line, 2p); PDT (1st line, 1p); RT (3rd line, 1p)	
10 [39]	1	1	2	NR	16		16		32		1		1 RC	NR	Yes (12)	No	NR	NA		PDT + cryosurgery (2nd line), RT (3rd line)	
	1	1	NR	NR	NR		NR		NR		1		1 RC	NR	Yes (24)	No	NR	NA		PDT + cryosurgery (2nd line)	
11 [1]	2 (10p)	2 (10p)	3 (10p)	NR	24 (10p)		24 (10p)		72 (10p)		1 (10p)		10 RC	NR	No: 10p	No (10p)	NR		0.1–6	Surgery (1st line, 10p) IMQ (post-surgery, 10)	
12 [14]	1 (5p)/3(1p)	1 (5p)/2 (1p)	3 (6p)	yes (2p)/no (4p)	16	8–16	16	8–16	48		24–48	1 (3p), 2 (2p), 3 (1p)	3 RC, 2 RP, 1 SD		NR	Yes:1p (18), No: 2p, NA: 3p	NA	soreness, Irritation, erythema (4p)	1.5	0–2	Surgery (2nd line)
13 [40]	2, 3 (2p)	3 (2p)	3 (2p)	No	12 (2p)		12 (2p)		36 (2p)		1 (2p)	2 RC	2 RC		No (2p)	No (2p)	mild itching	2 (1p), 3 (1p)		CO2 laser (2p) + cryosurgery(1p) (1st line); PDT followed by IMQ (2nd line, 2p)	
14 [33]	1 (6p)/3 (15p)	1 (6p)/≥2 (15p)	2, 3	Yes: stop due to side effect (1p)/application error (1p)	12 (14.7)	4–52	12 (14.7)	4–52	36		8–104	1 (11p), 2 (6p), 3 (2p)	11 RC/6 RP/2 SD		11 RC	Yes: 1p (23)	Yes (1p)	NR	NR	Surgery (before IMQ, 5p/after IMQ, 1p); CO2 laser (before IMQ, 3p); TFD (before IMQ, 1p)	
15 [34]	1 (7p)/3 (3p)	1 (7p)/≥2 (3p)	3	No	5	4–7	5	4–7	15		12–21	1 (9p), 2 (1p)	9 RC/1 RP		9 RC/1 RP	No	No	Irritation, erosion	1.5	0.1–3.5	Surgery (1st line, 3p)
16 [27]	3 (8p)	2 (8p)	3	Yes (1p): side effects	12		12		36		1 (6p)/2 (2p)	6 RC/2 RP	6 RC/2 RP		Yes: 4p (Md 4, 4–10)	Yes (3p)	Erythema, pain/burning	4	0.5–6	Surgery (1st line, 8p)	
17 ^4^ [3]	1 (4p)/3 (3p)	1 (4p)/3 (3p)	3 (6p)/7 (1p)	No	124	30–400	124	30–400	372		192–1200	1 (3p), 2 (4p)	3 RC/4 PR		2 RC	Yes: 3p (3–9)	Yes (3p)	Erythema (6p); pain (3p); hypopigmentation (2p); allodynia (1p), atrophy (1p), itch (1p), hyperpigmentation (1p), swelling (1p), weeping (1p), erosions (1p), edema (1p)	2.5	0.5–8	Surgery (1st line, 2p)
18 [29]	1 (5p), 3 (4p)	1 (5p), after surgery (4p)	3	Yes (3p): side effects	16 (8p), 6 (1p)		16	8–16	48		18–48	1 (5p), 2 (4p)	5 RC/4 RP	4 RC/3 RP	Yes: 3 (Md 36, 22–46)	No	irritation, erythema erosions (3p)	3.5	0.1–4		Surgery (before IMQ, 4/after recurrence, 1p)
19 [41]	2 (2p)	2 (2p)	7	No	12		12		84		1 (2p)	2 RC	1RC/1 NR		No	No	Burning (2p)	1 (2p)		PDT (before IMQ)	
					6		6		42												
20 [30]	NR	NR	1–5	NR	3 to 48		NR		NR		1 (4p), 2 (7p), 3 (4p), NR (3p)	1 (4p), 2 (7p), 3 (4p)	NR		NR	NR	NR	3 ^1^	0–37.5 ^1^	surgery (after IMQ, 4p)	
21 [15]	1 (20p), 2 (5p)	1 (20p), IMQ + IMb (4p), IMQ + 5-FU (1p)	3	NR		4–14		4–14			12–42	1 (1p), 3 (24p)	1 RC, 24 SD		1 RC	No	No	NR		1–10	Concomitant treatment: IMQ + IMB (4p), IMQ + 5FU (1p); Surgery (after topical, 18p)
22 [35]	1 (2p), 3 (4p)	1 (2p), 2 (3p), 3 (1p)	NR	NR	NR		NR		NR		1 (3p), 2 (2p)	3 RC/2 PR	2 PR		No	Yes (1p)	NR	5	2–8	Surgery (1st line, 3p; after IMQ, 2p); 5-FU (adjunctive to IMQ, 1p); Laser (adjunctive to IMQ, 1p); PDT (after IMQ, 2p)	
23 [28]	1 (19p), 2 (4p)	1 (19p), 2 (2p), 3 (1p), 2 (1p)	3	yes (8p; 34.8%): side effects	16 (21p), 11 (1p), 4 (2p)		16	4–16	48		12–48	1 (12p), 2 (7p), 3 (4p)	12 RC/7 RP/4 SD		10 RC/6 RP/4 SD	Yes: 8p (2p < 12; 6p Md 31, 14–46)	3p	pain/discharge and/or ulceration (79%); fatigue (67–71%); Headache (17–49%)	2.5	1–4	Surgery (1st line, 4p; after IMQ, 6p); IMQ (1p after surgery use IMQ, before the retreatment in this trial)
24 ^2^ [6]	1 (24p), 3 (31p)	1 (24p, 44%), ≥ 2 (31p, 56%)	2 (31p, 56%), 3 (24p, 44%)	Yes (4p stop IMQ): side effects	<36 (26 p, 51%), ≥36 (25p, 49%)		NR		NR		1 (22p, 43%)/2–3 (29p, 57%)		22 RC, 29 SD	22 RC	No	NR	erosions and local burning (2p), flu-like syndrome (2p)	(5.5) ^3^	1.5–12 ^3^	Surgery (1st line, 31p, 56%)	

^1^: This value refers to the total number of participants. The data relating specifically to the cases treated with imiquimod are not available. ^2^: Data and analysis of treatment management and outcome exclude four patients who did not tolerate the IMQ treatment. ^3^: This value refers to the complete response of patients. ^4^: In this article, there are 6 patients with 7 lesions; to facilitate the description of the data, the 7 lesions will be interpreted as 7 patients. IMQ (imiquimod); NR (no reported); NA (not applicable); CR (complete response); PR (partial response); SD (stable disease); PD (progression and increase disease); p (patients); 1st (first); 2nd (second); 3rd (third); IMb (ingenol mebutato); 5-FU (5-fluouracil).

**Table 2 cancers-15-05665-t002:** Outcome of imiquimod treatment in EMPD.

Author, Year	N°	Outcome								R	%	No R	%	Recurrence	%
		CR	%	PR	%	SD	%	PD	%						
Zampogna, J.C., 2002 [20]	2	2	100	0	0	0	0	0	0	2	100	0	0%	0	0
Mirer, E, 2006 [21]	2	1	50	1	50	0	0	0	0	2	100	0	0%	0	0
Hatch, K.D., 2008 [39]	2	2	100	0	0	0	0	0	0	2	100	0	0%	0	0
Challenor, R., 2009 [41]	2	2	100	0	0	0	0	0	0	2	100	0	0%	0	0
Tanaka, V.D.A., 2009 [35]	1	0	0	1	100	0	0	0	0	1	100	0	0%	NA	NA
Pang, J., 2010 [5]	12	6	50	6	50	0	0	0	0	12	100	0	0%	NR	NR
Sendagorta, E., 2010 [34]	3	3	100	0	0	0	0	0	0	3	100	0	0%	0	0
Shelbi, C.J.O., 2011 [45]	2	0	0	0	0	2	100	0	0	0	0	2	100%	NA	NA
Baiocchi, C., 2012 [34]	4	1	25	2	50	0	0	1	25	3	75	1	25%	1	100
Boulard, C., 2013 [42]	2	2	100	0	0	0	0	0	0	2	100	0	0%	2	100
Choi, J.H., 2013 [1]	10	10	100	0	0	0	0	0	0	10	100	0	0%	0	0
Sanderson, P., 2013 [14]	6	3	50	2	33	1	17	0	0	5	83	1	17%	1	33
Jing, W., 2014 [43]	2	2	100	0	0	0	0	0	0	2	100	0	0%	0	0
Luyten, A., 2014 [36]	21	11	52	6	29	2	10	0	0	17	81	2	10%	1	9
Marchitelli, C., 2014 [37]	10	9	90	1	10	0	0	0	0	10	100	0	0%	0	0
Cowan, R.A., 2016 [30]	8	6	75	2	25	0	0	0	0	8	100	0	0%	4	67
Liau, M.M., 2016 [3]	7	3	43	4	57	0	0	0	0	7	100	0	0%	3	100
Sawada, M., 2018 [32]	9	5	56	4	44	0	0	0	0	9	100	0	0%	3	60
Apalla, Z., 2018 [44]	2	2	100	0	0	0	0	0	0	2	100	0	0%	0	0
Van der Linden, M., 2019 [33]	18	4	22	7	39	4	22	0	0	11	61	4	22%	NR	NR
Choi, S., 2021 [15]	25	1	4	0	0	24	96	0	0	1	4	24	96%	0	0
Christodoulidou, M., 2021 [38]	5	3	60	2	40	0	0	0	0	5	100	0	0%	0	0
Van der Linden, 2022 [31]	23	12	52	7	30	4	17	0	0	19	83	4	17%	8	50
Borella, F., 2022 [6]	55	22	40	0	0	29	53	0	0	22	40	29	53%	0	0
Total	233	112	48	45	19	67	29	1	0	157	67	67	29	23	12

N° (number); CR (complete response); PR (partial response); SD (stable disease); PD (progression and increase disease); R (response).

**Table 3 cancers-15-05665-t003:** First-line treatment of Extramammary Paget’s disease.

i.d.	First Line Treatment					Adjunctive IMQ
	IMQ	Surgery	PDT	Laser	Other Treatments	
1 [19]	0	0	0	0	1	0
	1	0	0	0	0	0
2 [20]	1	0	0	0	0	0
	1	0	0	0	0	0
3 [36]	0	1	0	0	0	0
	0	1	0	0	0	0
4 [38]	0	1	0	0	0	0
	0	1	0	0	0	0
5 [32]	1	0	0	0	0	0
6 [5]	8	4	0	0	0	0
7 [34]	1	0	0	0	0	0
	1	0	0	0	0	0
	1	0	0	0	0	0
8 [42]	1	0	0	0	0	0
	1	0	0	0	0	0
9 [31]	1	2	1	0	0	0
10 [39]	1	0	0	0	0	0
	1	0	0	0	0	0
11 [1]	0	10	0	0	0	10 (surgery)
12 [14]	1	0	0	0	0	0
	1	0	0	0	0	0
	1	0	0	0	0	0
	1	0	0	0	0	0
	0	1	0	0	0	0
	1	0	0	0	0	0
13 [40]	0	0	0	1	0	1 (PDT)
	0	0	0	1	0	1 (PDT)
14 [33]	6	5	1	3	0	0
15 [34]	7	3	0	0	0	0
16 [27]	0	8	0	0	0	0
17 [3]	4	3	0	0	0	0
18 [29]	5	4	0	0	0	0
19 [41]	0	0	1	0	0	1 (PDT)
	0	0	1	0	0	1 (PDT)
20 [30]	NR	NR	NR	NR	NR	NR
21 [15]	20	0	0	0	5	5 (5-FU, IMb)
22 [35]	2	3	0	0	0	2 (laser, 5-FU)
23 [28]	19	4	0	0	0	0
24 [6]	24	31	0	0	0	0
Total	112	82	4	5	6	21

IMQ (imiquimod); PDT (photodynamic therapy); IMb (ingenol mebutato); 5-FU (5-fluouracil).

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
