# Peer review of "Topical Imiquimod in Primary Cutaneous Extramammary Paget’s Disease: A Systematic Review"

_cancers, 2023, doi:10.3390/cancers15235665_

Round 1
Reviewer 1 Report
Comments and Suggestions for Authors
Thanks a lot for allowing me to review this interesting paper. Vular Paget disease is a quite rare condition were surgical management often fails. For this reason the need for medical treatment is high.
The author review the literature on this topic but it should be accurately stressed that some studies report on results also on other site of disease outside the vulva and this may be a bias.
Moreover in many cases the schedule of the treatment was different.
Despite this limitations the paper covers a very intereesting topic and is worthy to be published.
Comments on the Quality of English LanguageGood paper on a rare but interesting topic. Worthy to be published.
Author Response
Thank you for your kind comment. Indeed, most of the publications of extrammamary Paget’s disease focus on vulvar Paget's disease, while this systematic review included extrammamary Paget’s disease localized in any anatomical area.
Reviewer 2 Report
Comments and Suggestions for Authors
Dear authors,
thank you for the interesting paper. I hope this will help to shed some light on an emerging treatment of a rare disease. The study design is appropriate and includes all the relevant studies on the topic. However some issues must be addressed before considering it for publication.
- as for the introduction, a recent multi society statement has been issued by a board of expert, recommending the use of imiquimod in vulvar Paget disease. I think it should be worth citing: 10.1136/ijgc-2021-003262
- line 64: correct "immunotherapy"
- line 71 onward: I think you should provide updated references upon that. I see that a recent review article on Imiquimod has been published, you should consider citing it: 10.1002/jmv.29238
- line 79: you should add reference 50 (10.1016/j.suronc.2021.101581) to this statement (as you later stated in line 355)
- line 217: correct "the majority.."
- line 325-326: I prefer to refer to non-preatreated EMPD as "naive EMPD"
- line 337 onward: uptake references for Imiquimod activity (10.1002/jmv.29238)
Thank you for your precious work
Comments on the Quality of English Language
Minor editing
Author Response
Thank you for your review and your suggestions. We add the interesting publications that you recommended. We have also made the suggested orthographic corrections.
Round 2
Reviewer 2 Report
Comments and Suggestions for Authors
Dear authors,
thank for your precious work. The manuscript has been well improved and deserves publication.
Congratulations